# Learning Representations that Enable Generalization in Assistive Tasks

Jerry Zhi-Yang He[1], Aditi Raghunathan[2], Daniel S. Brown[3],
Zackory Erickson[2], and Anca D. Dragan[1]

[1]University of California Berkeley, [2]Carnegie Mellon University, [3]University of Utah,
{hzyjerry,anca}@berkeley.edu, dsbrown@cs.utah.edu, {raditi,zerickso}@cmu.edu

**Abstract:** Recent work in sim2real has successfully enabled robots to act in physical environments by training in simulation with a diverse "population" of environments (i.e. domain randomization). In this work, we focus on enabling generalization in *assistive tasks*: tasks in which the robot is acting to assist a user (e.g. helping someone with motor impairments with bathing or with scratching an itch). Such tasks are particularly interesting relative to prior sim2real successes because the environment now contains a *human who is also acting*. This complicates the problem because the diversity of human users (instead of merely physical environment parameters) is more difficult to capture in a population, thus increasing the likelihood of encountering out-of-distribution (OOD) human policies at test time. We advocate that generalization to such OOD policies benefits from (1) learning a good latent representation for human policies that test-time humans can accurately be mapped to, and (2) making that representation adaptable with test-time interaction data, instead of relying on it to perfectly capture the space of human policies based on the simulated population only. We study how to best learn such a representation by evaluating on purposefully constructed OOD test policies. We find that sim2real methods that encode environment (or population) parameters and work well in tasks that robots do in isolation, do not work well in *assistance*. In assistance, it seems crucial to train the representation based on the *history of interaction* directly, because that is what the robot will have access to at test time. Further, training these representations to then *predict human actions* not only gives them better structure, but also enables them to be fine-tuned at test-time, when the robot observes the partner act. https://adaptive-caregiver.github.io.

**Keywords:** assistive robots, representation learning, OOD generalization

## 1 Introduction

Our ultimate goal is to enable robots to assist people with day to day tasks. In the context of patients with motor impairments, this might mean assistance with scratching an itch, bathing, or dressing [1, 2, 3]. These are tasks in which doing reinforcement learning from scratch in the real world is not feasible, and so sim2real transfer is an appealing avenue of research. Sim2real methods for physical robot tasks in isolation typically work by constructing a diverse "population" of environments and training policies that can work with any member of the population (e.g. a range of parameters of a physics simulator or a range of lighting and textures) [4, 5, 6, 7, 8, 9, 10, 11].

Similarly, population-based (self-play) training has proven successful in zero-sum games against humans [12, 13, 14]. But unlike tasks the robot does in isolation, assistance requires coordinating with a human who is also acting. And unlike competitive settings, assuming the human to be optimal when they are not, can result in dramatically poor performance [15]. Thus, in sim2real for assistance, we have to design a population of potential users and strategies to train with, akin to the physical environment parameters in typical sim2real tasks, rather than the standard population-based training approaches used in competitive settings. But designing a population that is diverse and useful enough to enable generalization to test-time humans, each with their own preferences, strategies, and capabilities, remains very challenging, making it likely that test-time partners might lie outside of the

6th Conference on Robot Learning (CoRL 2022), Auckland, New Zealand.

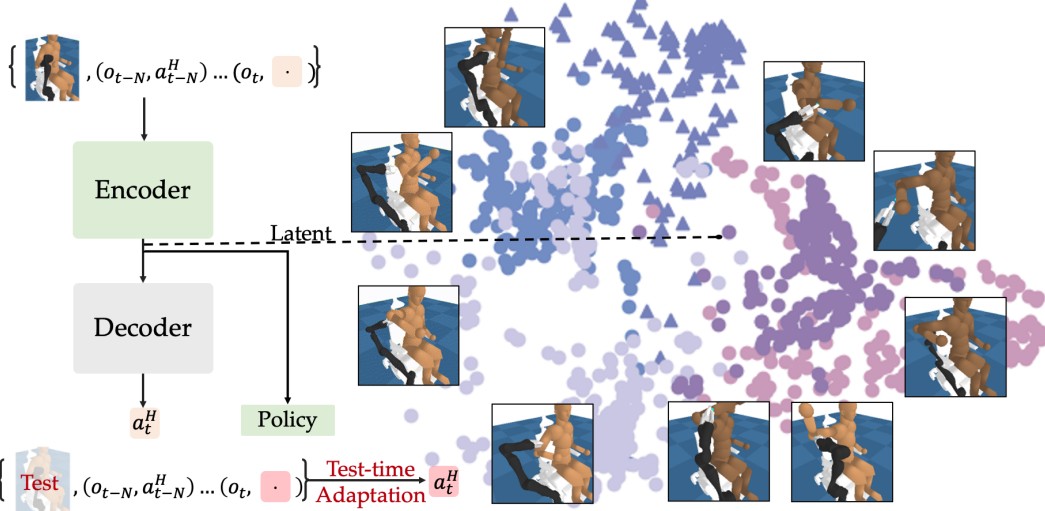

Figure 1: The framework for jointly learning the Personalized Latent Embedding Space and the robot policy. During training time, we train all components end-to-end to optimize for action prediction (orange) and the robot policy (green). At test time, we can further optimize for this objective to perform test-time adaptation (red). The resulting latent space captures the underlying structure of the preferences and strategies of the training humans.

distribution the population was drawn from. Therefore, sim2real methods for assistance will need to be ready to *generalize* to *out-of-distribution* partner policies.

In this work, we identify two principles as key to enabling better generalization. First is that we benefit from learning a latent space of partners that distills their policies down to a structure that is useful for the robot's policy *and* that makes it easy to identify partners at test time. Second is that we need to be prepared for this space to not perfectly capture the space of real human policies, and design it so that it is *adaptable* with real test-time interaction data.

We thus propose a framework that *learns a latent space directly from history of interaction by predicting the partner's actions*. Our framework allows a robot to capture the relevant information about the human partner that the robot can actually identify when starting to interact, and also enables test-time adaptation of the latent space itself when observing the partner's actions. When evaluated with partner policies we purposefully design to be out-of-distribution, we find that our approach leads to better generalization than prior methods which either do not learn a latent space at all [16], do not learn a latent space directly based on interaction history [7], or train a latent space based on other observables, like states or rewards [17, 18]. Our contributions are four-fold:

1. We introduce an assistive problem setting where the focus is explicitly on generalization to out-of-distribution partner policies.
2. We introduce a framework for training policies for this problem setting, Prediction-based Assistive Latent eMbedding (PALM). This enables us to study different methods for learning latent representations on how well they enable generalization.
3. We identify that the design choice of training a latent space by *predicting partner actions directly from history* outperforms (1) state-of-the-art sim2real approaches used in non-assistive tasks that are based on embedding environment parameters [7] as well as (2) human-robot interaction approaches that train representations by predicting observed states or rewards [17, 18].
4. We propose to adapt the learned latent space at test time, upon observing the partner's actions, and show it leads to generalization performance gains.

## 2   The Assistive Personalization Problem

In this section, we introduce the personalization problem in an assistive context. In particular, our goal is to learn a robot policy $\pi_R$ that can assist a novel human partner in zero-shot fashion, or with a small amount of test-time data.

**Two-player Dec-POMDP.** An assistive task can be modeled as a two-agent, finite horizon decentralized partially-observable Markov decision process (Dec-POMDP) and is defined by a tuple

$\langle S, \alpha, A_R, A_H, \mathcal{T}, \Omega_R, \Omega_H, O, R \rangle$. Here $S$ is the state space and $A_R, A_H$ are the human's and the robot's action spaces, respectively. The human and the robot share a real-valued reward function $R : S \times A_R \times A_H \rightarrow \mathbb{R}$; however, we assume that the reward function is not necessarily observed by the robot, i.e. its parameters (e.g. the location of the human has an itch) are in the hidden part of the state. $\mathcal{T} : S \times A_R \times A_H \times S \rightarrow [0, 1]$ is the transition function, which outputs the probability of the next state given the current state and all agents' actions. $\Omega_R$ and $\Omega_H$ are the sets of observations for the robot and human, respectively, and $O : S \times A_R \times A_H \rightarrow \Omega_R \times \Omega_H$ represents the observation probabilities. We denote the horizon of the MDP by $T$.

**Target User.** We target users with partial motor functions — a common impairment for individuals with partial arm functions. This is an impairment that can occur in some people with cervical SCI, ALS, MS, and some neurodegenerative diseases — leading to the need for robotic assistance. We model the extent of the impairment as the privileged information in the Dec-POMDP. The robot does not know this a-priori and thus needs to adapt to individual users' capabilities.

**The Robotic Caregiving Setup.** We define the observation space for the robot and the human following [3]: the robot observes its own joint angles, and the human's joint positions in the world coordinate and contact forces; the human observes their joint angles (proprioception) and the end-effector position of the robot. When training with simulated humans, the robot gets a reward signal (which depends on privileged information), and has to use that signal to learn to implicitly identify enough about the human to be useful; at test time, the robot does not observe reward signal and must use what it has learned at training time to identify the human's privileged information and be helpful.

**Distributions of Humans.** Let function $\pi_H : \Omega_H^* \times A_H \rightarrow [0, 1]$ be the human policy that maps from local histories of observations $\mathbf{o}_t^H = (o_1^H, \ldots, o_t^H)$ over $\Omega_H$ to actions. We define two distributions of human policies $\pi_H \in \mathcal{D}_{\text{train}}, \mathcal{D}_{\text{test}}$. In the assistive itch scratching, $\mathcal{D}_{\text{train}}$ can be a set of humans with different itch positions on their arms, which lead to their different movements. We refer to them as *in-distribution humans*. $\mathcal{D}_{\text{test}}$ contains *out-of-distribution* humans whose itch position differ from those in the $\mathcal{D}_{\text{train}}$. At training time, the robot has access to $\mathcal{D}_{\text{train}}$. Thus, it has ground-truth knowledge about the training human's privileged information, such as each human's itch position. At test time, we evaluate the robot policy by sampling humans $\pi_H \sim \mathcal{D}_{\text{test}}$ and directly pairing them with the robot policy. We evaluate the zero-shot and few-shot adaptation performance of the robot policy.

**Objective.** The main problem we study is how to leverage the training distribution to learn a robot policy $\pi_R : \Omega_R^* \rightarrow A_R$ such that we achieve the best performance on test humans. Concretely, we define the performance of the robot and human as

$$J(\pi_R, \pi_H) = \mathbb{E} \left[ \sum_{t=0}^{T} R(s_t, \pi_R(\mathbf{o}_t^R), \pi_H(\mathbf{o}_t^H)) \right], \tag{1}$$

Only given access to $\mathcal{D}_{\text{train}}$, our objective is to find the robot policy $\pi_R = \arg\max_\pi J(\pi, \pi_H), \pi_H \sim \mathcal{D}_{\text{test}}$.

## 3 Learning Personalized Embeddings for Assistance with PALM

In this section, we present Prediction-based Assistive Latent eMbedding (PALM). We introduce the general framework of using a latent space to perform personalization in an assistive context. We then highlight the advantage of action prediction in contrast to prior works. Finally, we describe how we can optimize PALM at test time to work with out-of-distribution humans.

### 3.1 Learning an Assistive Latent Space

Given a training distribution of humans $\mathcal{D}_{\text{train}}$ [1], we would like to learn a robot policy that can adapt to assist new users. To achieve that, a robot must learn to solve the task while efficiently inferring the hidden component that differs across humans. One natural way to do so is to *learn a latent space* that succinctly captures what differs across humans in a way that affects the robot's policy. When deployed on a test human, the robot infers this latent embedding and uses it to generate personalized assistance.

We denote the latent space as $z_t \sim \mathcal{E}_\theta(z; \tau_{1:t})$, where $\mathcal{E}_\theta$ encodes the trajectory $\tau$ so far and outputs latent vector $z_t$. The robot uses this latent space to compute its actions $a_t^R = \pi_R(o_t^R, z_t)$. We train

---

[1] we describe how we generate this distribution in Sec. 4.1

the base policy $\pi_R$ and the latent space encoder $\mathcal{E}_\theta$ jointly as they are interdependent [19] — better robot policy leads to different trajectories across humans, which in turn leads to distinguishing $z$. See Appendix D on training details. Ideally, we would like $z$ to capture sufficient information to differentiate the humans, similar to performing a "dimensionality reduction" on human policies $\pi_H$. We hereby introduce different objectives for learning such latent space, and how our method — learning by action prediction — makes a good fit for the assistive personalization problem.

## 3.2 How to Construct the Latent Space

**Prior work and limitations** LILI [17] and RILI [18] learn a latent embedding of the humans by predicting the next observations and rewards. While they have been shown to work in predicting and influencing human behaviours, both methods assume access to the reward function at test time, which we do not have access to in the assistive setting — we don't know a-priori the preference and needs of a new user. RMA [7] enables fast robot adaptation by learning a latent space of environment parameters, such as friction, payload, terrain type, etc. While it works for a single robot, it is unclear in human-robot settings, how we can encode human motions and preferences as environment parameters.

**Learning by action prediction.** Given history $\tau_{t-N:t} = \left((o^R_{t-N}, a^H_{t-N}), \ldots (o^R_t, \cdot)\right)$ of $N$ robot observation and human action[2] pairs, as outlined in Fig. 1, we embed this trajectory to a low-dimensional manifold and use it to predict $a^H_t$. The intuition is that if we are able to predict the next action by this human, we extract the sufficient information about the human's policy $\pi_H$. The latent vector $z$ is representative of the trajectory so far and indicative of the person's future actions. We do this by training a decoder $\mathcal{D}_\phi$ parameterized by $\phi$ to predict the next action from the encoder's output $z \sim \mathcal{E}_\theta(z; \tau_{1:t})$.

$$\mathcal{L}_{\text{pred}} = \min_{\theta, \phi} ||\mathcal{D}_\phi(z) - a^H_{t+1}||^2 + c_{\text{KL}} \cdot KL(\mathcal{E}_\theta(z; \tau_{1:t}) || \mathcal{N}(z)) \tag{2}$$

The encoder $\mathcal{E}$ is a recurrent neural network parameterized by $\theta$. Here the second term is a regularization term motivated by Variational Autoencoder [20, 21], that enforces the latent space to follow a normal prior distribution. This encourages nearby terms in the latent space to encode similar semantic meanings. In the context of assistive tasks, this helps us better cluster similar humans closer in the latent space, and we show a didactic example in Sec. 4.3.

## 3.3 Latent Space Adaptation at Test Time

At test time, as we work with a new user, we would like our encoding of the new user to match the true latent information, $z^*$ of that user. In other words, we would like to minimize $||\mathcal{E}(\tau) - z^*||^2$. Because we do not know about the new users a-priori, we can only optimize for this objective via proxy, which we refer to as *test time adaptation*.

Since the PALM latent space is based on action prediction, we can adapt it to a new user by further optimizing the latent space. Note that because Eq. (2) requires only observation-action data, we do not need any additional label to perform test time adaptation. More formally, we collect a small dataset of test-time interaction trajectories, $\tau$, and perform a few gradient steps to optimize both the encoder and decoder for Eq. (2): $(\theta, \phi) \rightarrow (\theta, \phi) - \delta \nabla_{(\theta, \phi)} \mathcal{L}_{\text{pred}}(\mathcal{E}_\theta, \mathcal{D}_\phi, \tau)$.

The idea of test-time optimization has been shown to improve perceptual robustness for grasping in sim2real research [22]. We follow a similar pipeline, where we can improve the latent encoding by collecting unsupervised action data from test users. Here the main difference is instead of perceptual differences, our goal is to reduce the domain gap on test users.

## 4 Experiments

In this section, we evaluate our method PALM (Prediction-based Assistive Latent eMbedding) in collaborative human-robot environments of varying tasks and varying populations of human models. In particular, we focus on the out-of-distribution generalization by constructing different forms of out-of-distribution populations. We focus on empirically investigating the benefits of learning a latent space, the effect of different kinds of prediction on learning a useful latent space, the properties of learned latent spaces, and the gains from test-time adaptation to humans.

---

[2]We do not assume access to the person's sensorimotor action (e.g. joint torques). We define human action as change in the person's Cartesian pose, which can be tracked externally.

## 4.1 Environments

Here we introduce two environments where we study assistive personalization. In both environments, the robot has to infer some hidden information from the human in order to successfully solve the task. Note that these are examples meant for demonstrating the effectiveness of the algorithm, and we do not claim to solve the full robotic caregiving problem.

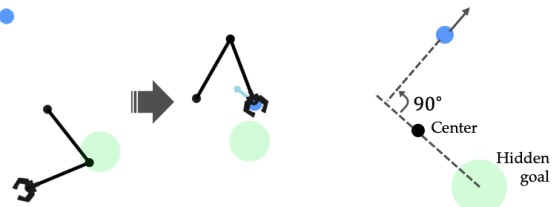

**Assistive Reacher** (Fig. 2) is 2D collaborative environment where a two-link robot arm assists a point human agent to get to the target position. This target is located at $(d\cos\alpha_\text{H}, d\sin\alpha_\text{H})$, where $d$ is a fixed value, and $\alpha_\text{H} \in [-\pi, \pi]$ is known to the human, but not the robot. The human agent is initialized randomly in the 2D plane with random hidden parameters $\alpha_\text{H} \in [-\pi, \pi], k_\text{H} \in [0.5, 1.5]$. The robot can only identify the target position by physical interactions — once the robot initiates contact, the human applies a force $k_\text{H} \cdot (\cos\alpha_\text{H} + \frac{\pi}{2}, \sin\alpha_\text{H} + \frac{\pi}{2})$. Only by recognizing the human in terms of $\alpha_\text{H}, k_\text{H}$ can the robot compensate the force, and successfully move the human to the hidden target. Each episode has 40 timesteps.

Figure 2: The assistive reacher environment. Left: the robot's goal is to move the human agent towards the hidden target. Right: the hidden goal's position can be inferred 90 degrees from the humans force output.

**The Scope of Generalization.** We define $\mathscr{D}_\text{train}$ as 36 samples uniformly sampled from $\alpha_\text{H} \in [-\pi, \frac{\pi}{2}], k_\text{H} \in [0.5, 1.5]$ and $\mathscr{D}_\text{test}$ as 12 samples uniformly sampled from $\alpha_\text{H} \in [\frac{\pi}{2}, \pi], k_\text{H} \in [0.5, 1.5]$.

**Assistive Itch Scratching** (Fig. 1) is adapted from assistive gym [3]. It consists of a human and a wheelchair-mounted 7-dof Jaco robot arm. The human has limited mobility — they can only move the 10 joints on the right arm and upper chest, and needs the robot's assistance to scratch the itch. An itch spot is randomly generated on the human's right arm. The robot does not directly observe the itch spot, and relies on interaction with the human to infer its location. Each episode has 100 timesteps.

We use co-optimization to create $\mathscr{D}_\text{train}$ and $\mathscr{D}_\text{test}$ for Assistive Itch Scratching. A benefit of the co-optimization framework is that it naturally induces reward-seeking behaviour from the human and the robot, which simulates assistance scenarios. For instance, to generate more inactive human policies, we can introduce a weighting term in the reward function for human action penalties $R_\text{p} = c_\text{p} \cdot ||\pi_\text{H}(s_t)||^2$ where $c_\text{p}$ is a constant controlling the penalty. The overall objective becomes

$$\max_{\pi_\text{H}, \pi_\text{R}} \mathbb{E}\Big[ \sum_t R\big(s_t, \pi_\text{H}(s_t), \pi_\text{R}(s_t)\big)\Big] + c_\text{p} \cdot ||\pi_\text{H}(s_t)||^2 \tag{3}$$

**The Scope of Generalization.** We are motivated by real world applications where users tend to have different levels of mobility limitations, or itch locations in different body parts. To generate a synthetic population to capture such diversity in itch scratching task, we explore different co-optimization settings (1) we assign different human action penalty to be $c_\text{p} = 3, 3.5, 4$, where larger penalties lead to the human agent exerting less effort. (2) We simulate different itch positions on the human's arm and train co-optimized human and robot policies conditioned on them. This leads to qualitatively different strategies for the human and the robot. Note that this serves first step to understanding how different methods generalize, since we never expect to be able to capture the diversity in

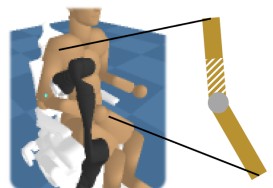

Figure 3: Definition of $\mathscr{D}_\text{train}$ and $\mathscr{D}_\text{test}$.

humans perfectly. For training, we use Proximal Policy Optimization (PPO) to optimize human and robot policies in an interleaving fashion. Note that we also keep the co-optimized robot policy and use it to obtain expert actions for assistive policy training (see Sec. 3.1 and supplement for full details).

To construct $\mathscr{D}_\text{train}$ and $\mathscr{D}_\text{test}$, we divide the two arms' areas into four equal portions, as shown in Fig. 3, and generate human policies conditioned on itch positions in these areas. $\mathscr{D}_\text{train}$ consists of three of the four portions and $\mathscr{D}_\text{test}$ consists of the remaining one. We then construct three distribution sets in increasing order of difficulties. In the first distribution $\mathscr{D}^1$, we confine itch positions from a line-shaped region. In $\mathscr{D}^2$, we sample from all the arm areas. Note that $\mathscr{D}^1$ and $\mathscr{D}^2$ are constructed by setting action penalty $c_\text{p} = 3$. In $\mathscr{D}^3$, we combine humans of $c_\text{p} = 3, 3.5, 4$, each trained with two

different random seeds. This adds the extra complexity of human activity levels. We simulate 12 in-distribution humans from each of the three training portions under each action penalty.

## 4.2 Baselines

We compare with baselines that do not learn an explicit latent space as well as existing methods for adaptation via learning latent embeddings.

**MLP and RNN.** We follow [16] that trains sequential models to enable adaptation to simulated humans. We explore using a recurrent neural network or a feed-forward network on concatenated state-action histories. The models directly output robot action, and there is no latent space modeling.

**ID-based Human Embeddings.** In contrast to learning latent space from history, another class of method studies encodes human-designed environment parameters [7, 19, 23]. We focus specifically on RMA [7], a two-phased method that first learns to encode task-ID (phase I) and then trains a recurrent network to regress to the embeddings from observation history (phase II). For training a quadruped robot, RMA encodes the physical parameters (friction, payload, etc) of the environment. The first stage trains a policy with ground-truth information, and the second phase performs environment identification. While RMA is shown to be effective for learning policy for in-distribution environments, it is unclear how well it generalizes to out-of-distribution environments. Furthermore, in assistive tasks, it is unclear how to construct the "ground-truth ID" for phase I that quantifies the user characteristics. We study **RMA-Param** and **RMA-Onehot**, where we assign each training human a one-hot vector. For RMA-Param, we use

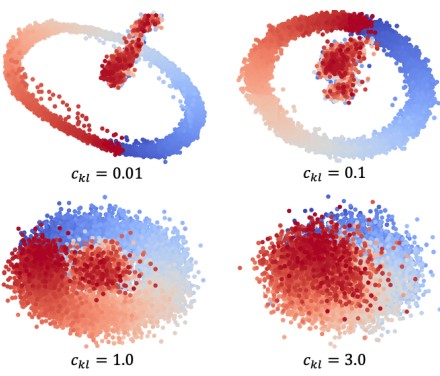

Figure 4: Latent space of PALM in the assistive reacher environment when we can sample humans $\alpha_H \in [-\pi \text{ (red)}, \pi \text{ (blue)}]$ continuously.

a three-dimensional vector that includes the $x, y$ position of the itch position and the arm index. When there are multiple human activity levels as mentioned in Sec. 4.1, we introduce a fourth dimension with an integer to indicate the action penalty $c_p$.

**LILI and RILI.** We consider two other methods of the PALM framework: LILI [17] and RILI [18]. As mentioned in Sec. 3.1, LILI jointly predicts future observation and reward, and RILI predicts reward. Given that reward is only available at training time, we cannot perform test time optimization for LILI and RILI.

**Ablations of PALM.** Our method has several components: we use a recurrent neural network to encode interaction history, and use its output to minimize prediction loss $\mathcal{L}_{\text{pred}}$ and policy loss $\mathcal{L}_{\text{pol}}$. We also use the KL term in Eq. (2) to regularize the human embeddings. To test the effectiveness of our method, we separate different parts and create a set of baselines. We hereby describe them in detail: (1) No $\mathcal{L}_{\text{pred}}$: the model shares the same encoder and policy network architecture, yet we don't optimize for $\mathcal{L}_{\text{pred}}$. By removing the prediction loss in Fig. 1, the latent space is not explicitly trained to contain human information. (2) No $c_{\text{KL}}$: no regularization in the latent space. (3) Frozen embedding: instead of jointly training embeddings and the policy network, we first train the encoder on expert data, freeze it, and then train robot policy. We include an ablation study of our main experiments in the appendix.

## 4.3 Didactic Experiment in Assistive Reacher Environment

**Can PALM learn a meaningful distribution from the interaction?** Unlike other ID-based methods like RMA, PALM does not have access to the human parameters at training time. We study whether PALM can learn a meaningful latent space without explicitly knowing this information. We sample training humans from $\alpha_H \in [-\pi, \pi]$ continuously. We train PALM with different amount of prior regularization, $c_{KL}$ from Eq. (2). We train using a recurrent window of length 4 and a batch size of 512 episodes. Additional training details can be found in the supplementary material.

We average test results using 100 episodes and visualize the results in Fig. 4.3. Given that humans are parameterized by $\alpha_H, K_H$, the ideal embedding space looks like a ring with a small blob in the center.

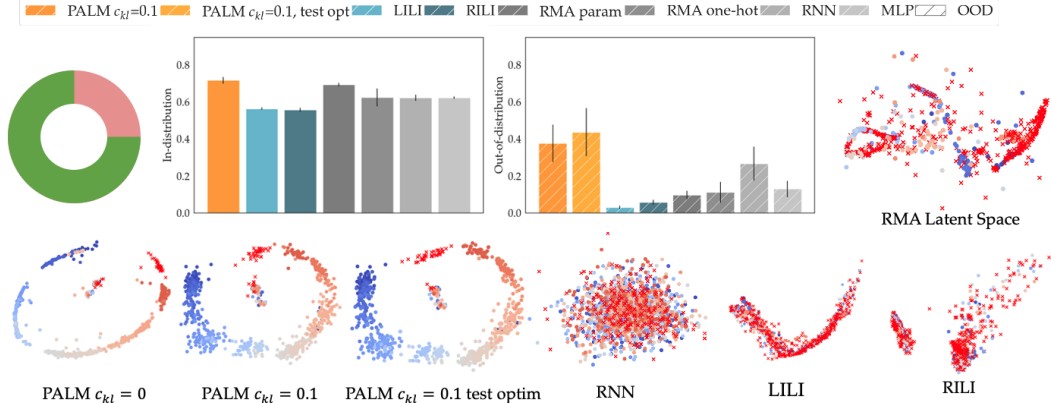

Figure 5: Top left: evaluation of PALM and baselines on in-distribution (green) and out-of-distribution (pink) humans. Right and bottom: visualization of the latent embeddings of different methods. OOD humans are highlighted in red crosses. Best viewed electronically.

The ring corresponds to the 2D projection of $\alpha_H$ and the blob denotes the initial part of interaction before contact, which is indistinguishable. We find that while PALM never observes the underlying parameter $\alpha_H$, it can learn a latent space that characterizes $\alpha_H$. Interestingly, varying the amount of regularization qualitatively affects the shape of the latent space. Setting the VAE regularization $c_{KL} = 0.1$ recovers a latent space that most resembles to the ideal latent space.

## 4.4  Assistive Reacher Main Experiment

**Experiment Setting.** We use the finite $\mathscr{D}_{\text{train}}$ described in Sec. 4.1 and train all baselines for 200 epochs with 512 batch size. We then evaluate the trained policies on $\mathscr{D}_{\text{test}}$. We normalize the resulting reward with respect to oracle reward.

**Results.** We average test results using 100 episodes. On in-distribution humans, we find that all methods successfully follow the right policy that assists the human to reach their goal. This shows that they all successfully predict the human latent information explicitly or implicitly. On out-of-distribution humans, the methods are no longer guaranteed to predict the correct embedding. PALM with action prediction significantly outperforms other methods. With test-time adaptation, PALM further improves.

**Visualizing the latent space.** We qualitatively study generalization by visualizing the latent space as well as the mapped embeddings of both in-distribution and out-of-distribution humans (in red crosses) in Fig. 5. Interestingly, only PALM with action prediction can infer the "ring" structure. RMA, RNN, LILI and RILI fail to do so. We hypothesize that because hand-crafted human IDs do not convey the information about human policy, RMA warped the IDs in arbitrary what that are harmful for generalization. The same happens with RILI and LILI. We hypothesize this is due to the inherent ambiguity in reward prediction: a low reward does not necessarily recover the human policy structure.

The visualization also offers some insight into why having $c_{KL}$ regularization is helpful for generalization. Compare the latent space of PALM $c_{KL} = 0$ and PALM $c_{KL} = 0.1$, the latter induces a smoother distribution where test humans are better fitting in the "missing arc" of the "ring". Further more, we see that with test time optimization, the PALM latent space embeds the OOD human better, by filling in more of the arc.

## 4.5  Assistive Itch Scratch Main Experiment

**Results.** We follow a similar procedure as the reach environment to train itch scratching policy, and train for 240 epochs with 192 trajectories for batch size. As shown in Fig. 6, we observe PALM with action prediction has better generalization performance than other baselines. We see that in the simplified distribution $\mathscr{D}^1$, RILI and MLP have the best generalization performance among baselines, yet as the complexity of the training human distribution increases, they deteriorate. Detailed results of the ablation study are included in the appendix.

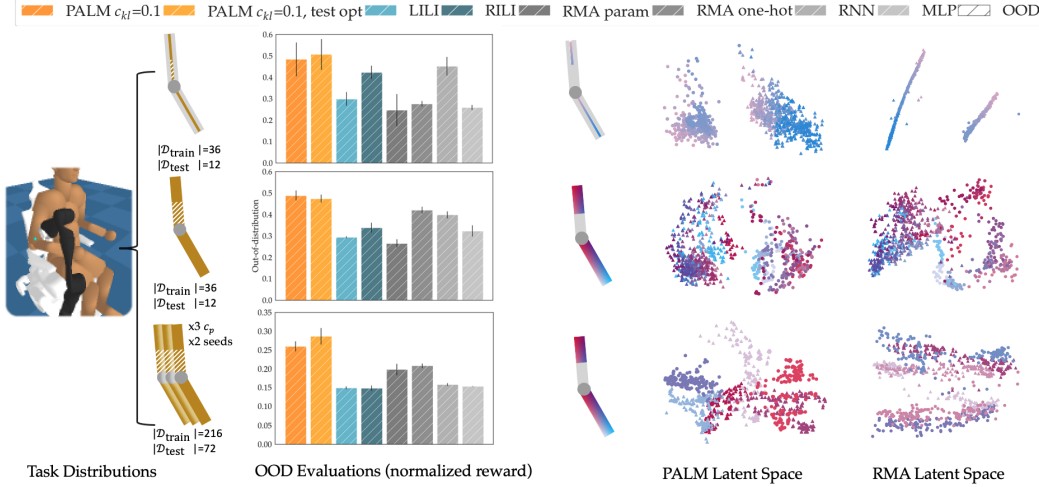

Figure 6: In assistive itch scratching, we sample humans by different itch positions and activity levels (varying action penalty $c_p$). We visualize the in- and out-of-distribution humans on the two-link arm figures. We also visualize the embedding space of PALM and RMA, where we color-code the embeddings of in-distribution humans. Here we leave the embeddings of other baselines to supplementary material.

**Visualizing the Latent Space** To further investigate why PALM generalize better to OOD human than RMA baselines, We visualize the latent space of the "straight-line" distribution. As we see in Fig. 7, PALM can capture the structure in human training distribution as two clusters, and also correctly embed the OOD humans distribution as a part of the upper arm distribution. RMA-based methods, on the other hand, can discover the structure of training humans. Yet qualitatively, they fail the correctly embed the OOD humans in proximity to the upper arm distribution.

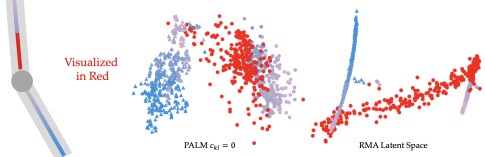

Figure 7: To better under extrapolations to OOD human, which have new itch locations highlighted in red on the left, we visualize the embeddings of both the IND (colored) and OOD (red) humans on the right.

### 4.6 Limitations and Failure Cases

Although PALM achieves good average-case performance, it works best with humans sampled near the training distribution. If we pair the robot with an adversarial human, PALM is likely to fail as it lacks a fall-back safety policy.

The major limitation of PALM is the requirement of generating a human population. While we provide one way to generate human populations based on weighted human-robot co-optimization, we lack ways to systematically generate diverse and realistic human motions. One important direction for future work is to incorporate real user data to create training populations. Improving the realism of the training human population is likely a crucial step to supporting transfer to real partners.

One future direction is extending to settings with one patient and one human caregiver. While our framework still applies, this leads to new challenges including (1) learning a joint or separate latent space for human patient and caregiver, (2) modeling a population of human caregivers for training in simulation, and (3) modeling communication between the human caregiver and patient.

## 5 Conclusion

Generalization is an important task for assistive robotics, and in this paper, we formulate a problem setting that focuses on Out-Of-Distribution users. To that end, we contribute a framework PALM for learning a robot policy that can quickly adapt to new partners at test time. PALM assumes a distribution of training humans and constructs an embedding space for them by learning to predict partner actions. We can further adapt this embedding at test time for new partners. Experiments show that PALM outperforms state-of-the-art approaches. We are excited by the potential of using PALM to enable robotic assistance in the future.

**Acknowledgments**

We would like to thank Ashish Kumar for discussions on the RMA method and Annie Xie for providing the implementation for LILI. We would also like to thank Charlie C. Kemp for feedbacks and insights on assistive tasks. This research was supported by the NSF National Robotics Initiative.

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
