# OpenReview forum: "Learning Representations that Enable Generalization in Assistive Tasks"
_robot-learning.org/CoRL/2022/Conference — CoRL 2022 Poster_

### Official Review · Reviewer_Up7D · 2022-07-26

**Originality:** Good
**Technical Quality:** Good
**Clarity Of Presentation:** Good
**Impact:** 3

**Recommendation:**

Weak Accept: I recommend accepting the paper, but will not argue for my recommendation if the majority of other reviewers have a different opinion.

**Summary:**

This paper proposes that explicit modeling of a latent space results in better performance with OOD human policies on assistive tasks. The PALM framework learns an encoder that maps a history onto a latent code that is the policy is conditioned upon. Experiments show that PALM policies lead to good performance on test human policies on an assistive reacher and assistive itch scratching task.

**Issues:**

Please see the weaknesses above. I would especially appreciate clarity regarding the fine-tuning method proposed.

**Quality Of The Limitations Section:**

Additional details required

**Reviewer Expertise:**

4: The reviewer is confident but not absolutely certain that the evaluation is correct

**Robotics Focus:**

Highly relevant to robotics but no hardware experiments

**Strengths And Weaknesses:**

Strengths:
- Assistive robotics is an important area and this paper is relevant to the community. Adapting to unseen users is essential and methods that generalize well would be of interest to researchers applying machine learning methods towards assistive robots and more generally, HRI.
- To my knowledge, the method is novel (up to the points below); I have not seen work in assistive robotics that adapt the test time latent space. The unseen reward setting is interesting.
- The experiments do show that learning a suitable latent space is important for performance and that test time adaption helps.

Weaknesses:
- The paper could better describe the method, particularly time-time adaptation. If my understanding is correct, the authors propose to optimize the latent space with specific test user data. It was unclear to me why it is necessary to adapt the entire latent *space*, rather than just the latent code $z$? If a good latent space is already learnt, why is there a need to further refine it? Specifically, how does training distribution $p_{train}$ differ from test distribution $p_{test}$ in a manner that refining the latent space becomes necessary?
- On a related note, it is unclear how PALM would scale to more complex settings. It seems like test time adaption of the whole space may require significant amounts of data and more than a few gradient steps? Also, with insufficient data, there is a risk of overfitting to specific scenarios (even with a specific test user)?
- It would also be helpful if the authors can clarify how such a system could be used in a deployed real-world system.
- Moreover, the experimental domains are relatively simple, e.g., compared to the lunar lander and driving domains in [17]. Can the authors provide some discussion around how the method scales? As stated above, it would be helpful to clarify how the test time adaptation was performed in the experiments.
- The paper would benefit from additional details and minor corrections to improve clarify:
  - line 92: $\mathbf{o}^H$ ($t$ subscript is missing?)
  - What is the episode length (time-steps) for both experiments? For each test user, how many episodes are tested and used for adaptation?
  - I could not understand Fig 6; can the authors provide a summary of what the dots represent and how they relate to the arm?
  - line 190: “reward-seek” -> “reward-seeking”?
  - Figs 4 and 5 seem to have $c_{kl} = 0.1$ for the first two algorithms? I believe the first bar should be  $c_{kl} = 0$?


**Summary Of Recommendation:**

Getting robots to behave appropriately when helping unseen users is an important topic and I would like to support this work. However, there are technical issues that give me pause and I am unable to recommend an accept at this point. I am willing to adjust my scores based on the clarifications.

-- Post Response --
I have updated my score to a Weak Accept based on the clarifications.

---

> ### Author Response · Authors · 2022-08-26
> **Added details on test-time optimization; moved user study from appendix to main text**
>
> > Q: Discuss test time optimization. Why not use update z, but the entire latent space?
>
> Re: We thank the reviewers for the question. Because our method (as shown in Figure 1) encodes trajectory snippets of length N, there is no explicit enforcement that each person will always map to the same z. In fact, z is a representation of both the person and the current state of assistance. Because of that, we are not able to only update z, as it changes every timestep. By optimizing for encoder and decoder altogether, we can ensure that the test-time optimization has an effect on every timestep.
>
> > Q: If a good latent space is already learned, why is there a need to refine it further? How is training/test distribution different that test-time optimization is necessary?
>
> Re: We thank the reviewer for this clarification question. In our work, we define OOD distribution as humans that exhibit behaviors that are different from the training distribution. For example, in the assistive reacher environment, the OOD humans have new force directions; in the itch-scratching environment, the OOD humans have new itch locations, which lead to different behaviors. Because the robot has never seen such behavior in training time, but only similar ones, test-time optimization is necessary to help the latent space extrapolate to new humans.
>
> > Q: How much data is required for test-time optimization? Risk of overfitting?
>
> Re: This is a great clarification question. We have updated appendix section E to include more details about test-time optimization. In summary, we roll out the trained robot policy with the same user for 25 iterations, which amounts to 150 seconds of wall clock time. We then optimize for the prediction loss (including KL regularization term) using a learning rate of 0.0001 for one to five steps, and use the one with the lowest loss. We automatically tune the hyperparameters by using the training human distribution, where we collect a mini training set and evaluation set both of 25 iterations. We use the mini training set to tune the learning rate and use the mini evaluation set to ensure there is no over-fitting.
>
> > Q: How does it scale? It seems like test time adaption of the whole space may require significant amounts of data and more than a few gradient steps?
>
> Re: As discussed in the previous question, we experimentally find that our method does not require a significant amount of new data to adapt to OOD humans in assistive reacher and assistive itch scratching. One important requirement as we discussed in limitation (section 4.7) is the diversity and realism of training human distribution, which is a prerequisite for test-time optimization to work. Designing and learning better training human distributions is an exciting area of future work.
>
> > Q: Discuss deployment in real system
>
> Re: We thank the reviewer for this question as we believe that real-world application is essential for assistive robots. While we previously included a human user study in the appendix due to the page limit, we have moved it to the main text in this revision in section 4.6. We find that robot policy trained with PALM achieves good zero-shot generalization performance when interacting with real human participants.
>
> > Q: Too simple compared to lunar lander/driving. How does the method scale?
>
> Re: Thanks for the great question. We note that the Lunar lander has an 8-dimensional continuous observation space with 4 discrete actions. In comparison, the itch-scratching environment we use in our paper has a 24-dimensional continuous observation space with a 7-dimensional continuous action space for the robot. We believe that the high-dimensional continuous observation and action spaces make this domain challenging. We would also like to note that the lunar lander/driving environments in [17] have a different setup than ours. In [17] the authors make the assumption that the robot observes the reward signal of the other agent. Furthermore, in some of the environments in [17] the ego-agent cannot directly observe the other agent it is interacting with. This differs from our problem setting in which we assume that the human’s reward is unobservable but that we can observe the human’s behavior and adapt accordingly to assist the human. If the reviewer would like to see experiments in lunar lander/driving, we are happy to add them to the revised paper.
>
> > Q: Figs 4 and 5 seem to have ckl=0.1 for the first two algorithms? I believe the first bar should be ckl=0?
>
> Re: Thanks for the clarification. The first bars in figs 4 and 5 actually both correspond to ckl=0.1. The difference between the first and the second bar is that the second bar showcase the policy performance after test-time optimization.

---

### Official Review · Reviewer_EjUi · 2022-07-30

**Originality:** Fair
**Technical Quality:** Good
**Clarity Of Presentation:** Good
**Impact:** 3

**Recommendation:**

Weak Reject: I recommend rejecting the paper, but will not argue for my recommendation if the majority of other reviewers have a different opinion.

**Summary:**

The paper focuses on learning latent representations for generalization in assistive tasks such as reaching and itch-scratching. The latent space is learned directly from the history of interactions by predicting partner actions during training. During testing, the learned space is adapted by observing the partner actions to account for OOD partner policies.

**Issues:**

See the “Suggestions for Improvement” section above.


**Quality Of The Limitations Section:**

Additional details required

**Reviewer Expertise:**

5: The reviewer is absolutely certain that the evaluation is correct and very familiar with the relevant literature

**Robotics Focus:**

Relevant but unlikely to deploy to hardware in near future

**Strengths And Weaknesses:**

Strengths:

- The paper considers learning latent space representations from history of human actions that are observable during assistance, instead of states / rewards / environments
- The paper compares their methods with multiple baselines with different strengths

Suggestions for Improvement:

- The paper can be significantly improved by being more cohesive about the motivation, the claims, and the assumptions of the methods. The paper motivates itself by citing the use-case of robotic caregiving. Robotic caregiving is a multifaceted and highly contextual domain. No robotic caregiving context is complete without specifying the four things: 1) target user, 2) the caregiver (optional), 3) the environment, and 4) the robot. Without knowing the target user of this technology, it is unreasonable to assume that the user is able to move their arms during the activity, which affects the POMDP formulation, choice of action space, as well as the co-optimization framework that induces reward seeking behavior from the humans. If a patient could move their arms, can the paper justify why they would need robotic assistance with itch scratching. Without knowing the environment and the robot, it is unreasonable to assume that partner interactions themselves would be able to capture the differences between training time and test time for generalization. The paper needs to clarify the setting - if it is a robot-arm mounted on a wheelchair, are they considering generalization to a different user who is lying on a hospital-bed for itching? The assumptions need to be very clearly specified in the paper and the claims should be carefully considered and re-worded.

- There is a lack of real-world experimentation for these methods, which can provide more insights on the practicality of these approaches for the robotic caregiving context. Is this a realistic assumption that during training time the privileged information would be available for robotic caregiving tasks? Can the paper discuss how much data would be needed to train an encoder that takes in interaction data to generate latent space, and if that scale of robotic caregiving data collection is feasible in the real world? How would these methods translate beyond simulation?

- The paper can also be stronger by clarifying what personalization means in the context of learned embeddings (Section 3). Personalization in HRI and in the robotic caregiving world can have many meanings and it would make the paper clearer, if it can explain clearly in what context this is used.

- For itch scratching, can the paper justify why the action space only considers joint and end-effector positions? Without contact forces, how would itch scratching be possible to generalize between different users?

- How would this framework generalize when there is a human caregiver present to help the robot with the care recipient (two humans - one robot)? This is a very probable scenario for robotic caregiving (in fact more probable than just a dyadic robot-user scenario presented in this paper) and a discussion on extending the framework from a two-player Dec-POMDP would make the paper stronger.

- The regularization term (eq. 3) is motivated by VAE formulation and helps in grouping similar humans closer, assuming a normal distribution. By similar humans, the paper probably means similar itch locations and history of actions ( a clarification would be helpful). Have the authors considered other formulations that directly attempt to use VAE structures for sequential data such as Variational RNNs ( https://proceedings.neurips.cc/paper/2015/hash/b618c3210e934362ac261db280128c22-Abstract.html )? A discussion on the modeling choice would be helpful.

- Can the paper clarify how behavior cloning is used to stabilize their policy training? How is covariate shift dealt with in this case?

- The paper should be carefully revised for typos, punctuation, other formatting etc. Some examples:
     - Line 36: comma after when they are not
     - k_H should be defined in Section 4.1
     - Fig. 2 can use a more detailed caption.
     - Is the avatar in the assistive gym a female avatar? Kindly clarify the choice of pronouns (Line 84)


**Summary Of Recommendation:**

The paper in its current state needs realistic evaluations and justification of the choice of methods.

##########################
After rebuttal period:
##########################

I thank the authors for engaging in thorough discussions with me and responding to many of my comments. I really appreciate it. I have read the rebuttal in detail and I am revising my decision from "Strong Reject" to "Weak Reject". While I am still not fully convinced that the paper is showing evidence of what it is claiming which is to learn representations that enable generalization in assistive tasks, the paper has improved through these revisions. I have updated my scores accordingly. A side note: I understand that recruiting people with disabilities is indeed hard, but one thing to consider in the future would be to engage the stakeholders (occupational therapists, caregivers, care recipients etc.) even through some remote zoom discussions, atleast to confirm that the formalization and the experimental design is real-user inspired. Also, while experimenting with different activity levels is great, the mobility limitations of end users do not vary globally but are instead a function of relative local joint constraints which can be characterized by their AROM and PROM. Thank you for your work.

---

> ### Author Response · Authors · 2022-08-25
> **We addressed clarification questions, moved user study from appendix to main text, and added additional result for Variational RNN baseline**
>
> > Q: “The robotic caregiving setup is unclear”.
>
> Re: We thank the reviewer for pointing out the need to improve the clarity of the caregiving setups. We have updated section 2 and section 4.1 to describe the robotic caregiving setup, and more specifically the itch-scratching task. To respond to the reviewers' questions, we fully follow the setup in the assistive gym paper [1] for the robot and environment, where the patient is sitting in a wheelchair with a side-mounted Jaco robotic arm. The person has only partial motor function in their right arm—a common impairment for individuals with cervical spinal cord injury and some neurodegenerative diseases—leading to the need for itch scratching assistance. As now emphasized in the paper, we focus on user generalization over different user motor impairments — the users are at the same sitting pose, but have different capabilities to move their joints, which lead to different motions.
>
> > Q: lack of real-world experimentation.
>
> Re: We thank the reviewer for the feedback and we agree that real human experiments are important. In our submission, we included a study with real users in the appendix due to the page limit. We have moved it to the main body in section 4.6. We find that robot policy trained with PALM achieves good zero-shot generalization performance to real human participants.
>
> > Q: Personalization in HRI and in the robotic caregiving world can have many meanings. Clarify what personalization means in the context of learned embeddings.
>
> Re: We agree with the reviewers that the original wording is ambiguous. By personalization we mean to say adaptation, where the robot can achieve good assistance with new users in a zero-shot fashion, or with a small amount of test-time data. We have updated this in the first paragraph of section 2. We also added more details about test-time optimization for PALM in section E in the appendix.
>
> > Q: For itch scratching, why no contact forces?
>
> Re: We follow the exact setup in the assistive gym paper [1] and do consider contact forces at the robot’s end effector in its observation space. We have made this more clear in section 2 “The Robotic Care-giving Setup”.
>
> > Q: What if there’s a human caregiver?
>
> Re: We thank the reviewer for this interesting extension. We have added a note about this area of future work at the end of the paper in section 4.7. We think that the setting where there is a human caregiver and robot assisting a patient fits into our theoretical framework—our problem becomes a 3-player dec-POMDP, the human caregiver would be modeled as another partially observable agent and the robot could learn a latent space embedding the trajectories of both humans. This leads to new challenges including (1) whether to learn a joint latent space or independent latent spaces per human, (2) how to model a population of human caregivers for training in simulation and (3) how to model communication between the human caregiver and patient. We think this is an exciting area for future work.
>
> With that said, there is also value in investigating robotic caregiving in a scenario where no human caregiver is available. As populations age, there are often care settings with more patients than caregivers and so it can be important for robots to provide care even when a human caregiver is not immediately available.
>
> > Q: Have you considered other VAE formulations?
>
> Re: While we use the vanilla VAE definition [2] in this paper, our framework allows incorporating other formulations of sequence generative models. Developing even better models for human embedding is a future direction. We have also experimented with the paper [3] listed by the reviewers and added the results to section C in the appendix.
>
> > Q: How is behavior cloning used to stabilize training? How about covariate shift?
>
> Re: We thank the reviewer for the inquiry and have added more clarification in section 3.1. Our main motivation for training our final PALM policy with behavior cloning is the challenge of running RL on large human populations. Because we train with synthetic humans, we first optimize the expert robot policy for each human actor. We then query the expert for on-policy actions, as proposed by DAgger [4]. This design leads to no covariate shift for the training distribution.
>
> Citations
>
> [1] Erickson et al, Assistive Gym: A Physics Simulation Framework for Assistive Robotics, 2019
>
> [2] Bowman et al, Generating Sentences from a Continuous Space, 2015
>
> [3] Chung et al, A Recurrent Latent Variable Model for Sequential Data, 2015
>
> [4] Ross et al, A Reduction of Imitation Learning and Structured Prediction to No-Regret Online Learning, 2010

---

> > ### Comment · Reviewer_EjUi · 2022-08-27
> > **Follow-up questions**
> >
> > Thank you for taking the time to respond to my questions. I really appreciate it. However, I am still not convinced either by the setup, the human study, or how realistic the assumptions are. My biggest concern with this paper is the framing and focus on learning representations for generalization in assistive tasks, which seems a bit disconnected from understanding the nature of assistive tasks themselves and their challenges. More details below:
> >
> > 1. "The person has only partial motor function in their right arm—a common impairment for individuals with cervical spinal cord injury and some neurodegenerative diseases—leading to the need for itch scratching assistance".
> >
> > Not all individuals with cervical spinal cord injury and neurodegenerative diseases have partial motor functions. The paper should clarify who the specific target population is, that would need help with itch scratching and reaching, but still have partial motor functions. Clarifying this will make the paper grounded to realistic scenarios.
> >
> > 2. "lack of real-world experimentation with the target population".
> >
> > The pilot human user study was done with people with no mobility limitations holding HTC VIVE and moving. Can the paper show some rationale as to why these results should generalize to people with mobility limitations (which is the goal of the paper)? The population with whom the study was done and the target population would have very different motor functions and movements, and it is not clear if the assistance methods would transfer.
> >
> > 3. “Clarifying the setting”
> >
> > The paper needs to clarify the setting - if it is a robot-arm mounted on a wheelchair, are they considering generalization to a user who is lying on a hospital-bed for itch scratching? Or generalization to a different body part for itch scratching (I know that they are doing different itch points)? If not, they should specifically mention that the task is not “itch-scratching” but “itch-scratching on a wheelchair of (a) certain body part(s)” and similarly for “reaching”. The paper would become stronger if it is clarified what generalization (which is the goal of the paper) means in these contexts and how the scenarios chosen for generalization are inspired by real-world challenges of assistive tasks.
> >
> > 4. "Relevant data availability in the real world".
> >
> > Can the paper discuss how much data would be needed to train an encoder that takes in interaction data to generate latent space, and if that scale of robotic caregiving data collection is feasible in the real world? Note, the avatars and environments in the assistive gym are very different from a realistic assistive home environment with human avatars with mobility limitations.
> >
> > 5. “With that said, there is also value in investigating robotic caregiving in a scenario where no human caregiver is available.”
> >
> > I completely agree with the authors on this. However, for these physically assistive tasks, it may also be unreasonable to assume that that is the only viable option and therefore, a flexible framework with both two-player and three-player dec-POMDP would be more realistic.

---

> > > ### Author Response · Authors · 2022-08-27
> > > **Clarification on generalization; response to follow-up questions**
> > >
> > > We greatly appreciate the reviewers for the response! To our understanding, the core of the clarification questions is: to what extent is the method able to generalize, be it different body parts, different user mobility or different user poses?
> > >
> > > In this paper, we specifically study two specific cases of generalizations (section 4.1 “The Scope of Generalization”): users with different activity/impairment levels, and users with itch locations in different body parts. We fully agree with the reviewer that our experimental setup may be an instance of the broad robot caregiving application. We have narrowed down our claims to emphasize our exact settings (section 4.1).
> > >
> > > We also thank the reviewer for elaborating on the questions. Here are our responses:
> > > 1. We have updated our submission to clarify that the intended population of the assistive task is individuals with partial arm functions. This is an impairment that can occur in some people with cervical SCI, ALS, MS, and some neurodegenerative diseases.
> > > 2. The goal of our user study is to showcase that PALM — when trained on a synthetic population of humans — can assist real users with the same level of mobility. This showcases the possibility of using sim2real learning methods for assistive tasks, given the perceived difference between synthetic and real humans. This is more challenging than demonstrated in previous studies [18]. We agree with the reviewer that it is important to demonstrate with patients. Given the difficulty of recruiting users with mobility limitations, we look to study it in future work.
> > > 3. We have updated section 4.1 to emphasize that we are motivated specifically by (1) people with different levels of mobility limitations who can have different activity levels, and (2) people who have itch locations in different body parts.
> > > 4. Based on our training details (Appendix section E), our method takes 1,382,400 timesteps to train, which amounts to 23 hours of wall clock time interactions. We agree with the reviewer that in real life there are additional complications such as clothing, furniture, etc. We leave the problem of environment modeling and domain randomization to future work.
> > > 5. We agree with the reviewer that a fully flexible framework would be ideal and that our setup may not be realistic. However, we believe it is more grounded and realistic than prior work [18], which assumes one robot assistant and one user operating in a turn-based manner.

---

### Official Review · Reviewer_tdQt · 2022-07-31

**Originality:** Good
**Technical Quality:** Good
**Clarity Of Presentation:** Very Good
**Impact:** 3

**Recommendation:**

Weak Accept: I recommend accepting the paper, but will not argue for my recommendation if the majority of other reviewers have a different opinion.

**Summary:**

This paper addresses the problem of sim2real in the context of assistive tasks, in which there is an agent that is acting and requiring assistance.  If the diversity of behaviors is not properly captured, trained agents may encounter out-of-distribution policies at test time.  The authors posit that the robot will have access to interaction history at test time, which motivates learning a “good” latent parameterization that generalizes to humans and which are adaptable with test-time interaction data.  The authors offer empirical evidence to support that the latent representation may be interpreted and leveraged using human policies trained from a policy distribution, and that if out-of-distribution humans are encountered, the parameterization can be adapted accordingly.  The supplement and video also provides further experimental studies, including physical human-robot experiments.

**Issues:**

1. The authors are strongly encouraged to add a description of why other approaches (LILI and RILI) are not able to handle the assistive domain, and what aspects of the proposed approach lend to better identification of a "more correct" latent representation than these other approaches.
 - Does this relate at all to any unstated definitions or assumptions for $D_{train}$ and $D_{test}$?  For instance, are there limitations to the domain shift from $D_{train}$ to $D_{test}$ or can the shift be arbitrarily large?
 - The authors should also highlight any such differences in the experimental comparisons of Sec. 4.

2. There are missing details in Sec. 3.2:
  - Please explain why $\phi$ is not included in the test-time adaptation update.
  - Please explain how catastrophic forgetting is dealt with in the adaptation procedure.
  - Also, how does this factor into the selection of the hyperparameters such that the distribution shift to the test distribution is properly handled by the adaptive procedure, yet does not destroy the ability to assist within the original training distribution.

3. The authors should explain whether their approach in general can handle non-rational or suboptimal human participants, or whether any extensions are necessary.

4. Please fix the formatting issues.  There are not many, but one example is Figure 5, which is dislocated from where it is first referenced.

**Quality Of The Limitations Section:**

Additional details required

**Reviewer Expertise:**

4: The reviewer is confident but not absolutely certain that the evaluation is correct

**Robotics Focus:**

Sufficient demonstration on hardware

**Strengths And Weaknesses:**

Better computational approaches to solving the problem of human-robot cooperation and assistance is important for the robotics community in general.  In many studies, the human is assumed to be an adversary, where the robot must exploit the learned human strategy.  In this paper, the authors provide a solution where robot must assist the human in assistive HCI settings, such as itch scratching.  The high-level problem statement - a latent representation amenable for interaction, and which can be adapted to fine-tune the latent embedding according to any distribution shifts seen at test time - is a very important and impactful problem.

The paper is well-written, and comprehensive.  The authors provide reasonable context, and demonstrate, through compelling examples, the PALM approach, which illustrates the overall utility and provide a basis for comparison with other methods.  The comparisons between other approaches (RMA, LILI, RILI) strengthen the paper’s arguments.  The authors also provide (in the supplement and video) a real user study to further demonstrate their approach.

One of the weaknesses of this paper is that the link between how the proposed computational approach addresses generalizability is not clear.  After reading the paper several times, I could not figure out what aspects of the approach allow for a latent structure that is amenable to assistance.  Does the human have to behave optimally and rationally for the approach to work?  Relatedly, the term “generalizability” seems to be a misnomer in the context used in the paper.  If $D_{train}$ and $D_{test}$ are sufficiently different (by way of counterexample, suppose that $D_{test}$ leads to adversarial policies), then the learned structure may be completely useless at test time.

Also, a complete study of the generalizability of the approach was never fully explored, probably since it is not required for the assistance domain, since it is assumed that humans are mostly cooperative.  Could the authors adopt a more specific definition for $D_{train}$ and $D_{test}$, or provide assumptions on the nature of these distributions or the nature of the human behaviors, to make the assistance and generalization claims clearer?

Related to the points above, in the Experiments section, it is never explained why the results of the other methods (LILI, RILI, RMA) do not generalize well, yet PALM does.  Particularly, the results for the reacher domain show that the circular latent parameterization correctly emerges for PALM, even without test-time optimization, though this fails to emerge in the other approaches.  From a computational standpoint, is it one of the loss components (e.g. the PPO or BC loss components), or is it differences in decoder output or less reliance on rewards that causes PALM to predict a human better than LILI or RILI?

In Sec. 3.2, why does the adaptation step only update $\theta$ (and not also $\phi$)?  Clearly, the decoder was trained on $D_{train}$, and will suffer the same domain-shift issues as the encoder when presented with $D_{test}$?

Another significant weakness is that, in Section 3.2, the presented adaptation scheme is not at all convincing for solving the objective in eq. (2).  How does the approach deal with catastrophic forgetting of the original latent space?  How many iterations does there need to be for the test distribution to be accurately captured?  Since the solution to (2) cannot really be achieved (since there are no proofs to this effect), then this is entirely a heuristic / empirical claim.  What are the hyperparameters (e.g. $\delta$), and how are they selected?

In the Assistive Itch Scratching domain, the co-optimization of $\pi_R$ and $\pi_H$ seems limiting to implement in practice, since there are no ground truth rewards and humans could behave unlike eq. (5).  How would the approach work if the ground truth human policy $\pi_H$ was not co-trained?

There are various formatting issues.  For instance, Figure 5 (on p. 8) is dislocated from where it is first referenced (p. 6).

**Summary Of Recommendation:**

The proposed approach for learning the right representation for human-robot interaction in assistance settings is an important and sometimes overlooked area.  While this paper does not seem to propose a novel learning paradigm, it does focus on setting up the problem such that a coherent latent representation can be learned, and seems to generalize, to an extent, other similar approaches (LILI, RILI).  I believe that the proposed adaptation scheme does not solve the overall objective in eq. (2), and should be re-written.  However, I also feel that the paper carries strong empirical evidence pointing to promise of using such approaches in realistic settings.  My recommendation leans toward acceptance.

---

> ### Author Response · Authors · 2022-08-26
> **Addressed clarification issues on generalizability, test distribution; discussed challenges and limitations of co-optimization.**
>
> > Q:  How the proposed computational approach addresses generalizability is not clear.
>
> Re: We thank the reviewer for this clarification question. The key for PALM to achieve generalization is through building a latent space that captures the key aspect of humans in the training distribution (in other words, performing “dimensionality reduction”), and being able to map the test humans to these principle components of human policy. Our framework does not assume nor require that humans act rationally. Rather, we assume that we have a training distribution that captures similar human behaviors. That being said, humans tend to have various kinds of suboptimal biases, and constructing such training distribution poses a new challenge, as we discuss in section 4.7. One promising direction is to leverage real-world motion data to construct training humans that capture such characteristics.
>
> > Q: If train and test distribution are sufficiently different, then the learned structure may be completely useless at test time.
>
> Re: We fully agree with the reviewer that an “adversarial” human policy can render arbitrarily low performance for the robot policy. In fact, this is a known issue for deep learning-based robot policies. We believe that safeguarding against adversarial policies is a very important direction for improving our work.
>
> > Q: Could the authors adopt a more specific definition for Dtrain and Dtest ... to make the assistance and generalization claims clearer?
>
> Re: Thank you for the clarification question! For the itch-scratching domain we studied, we detailed the definition of Dtrain and Dtest at the end of section 4.1. In summary, the test humans have different itch locations from those in the training set, which renders qualitatively different behaviors as they tend to expose itch locations to the robot. On the other hand, the test humans share the same action space and reward function as those in the training human, which renders the same activity level.
>
> > Q: Why other methods (LILI, RILI, RMA) do not generalize well, yet PALM does.
>
> Re: Thanks for the great question!  We believe that the differences come from the formulation and policy structure of these methods. As we discussed in section 4.4, we hypothesize the reason for LILI and RILI to underperform is their reliance on the reward at test time. We also discussed in section 4.5 the possible failure reason for RMA, where it fails to embed test humans in proximity to the upper arm distribution. We think this is likely because the latent space of RMA relies on regression to handcrafted human IDs, which limits the generalizability. In terms of training loss, we use the same PPO + BC loss for all methods.
>
> > Q: Why only update theta, not phi? The decoder also has domain-shift issues right?
>
> Re: We thank the reviewers for this great clarification question! For test time optimization, we update all the components in Fig. 1 in an end-to-end fashion, which includes both the encoder and the decoder. We have updated section 3.2 to reflect this.
>
> > Q: Catastrophic forgetting of the original latent space? What are the hyperparameters?
>
> Re: Thank you for the question! We have updated appendix section E to include more details. In summary, we roll out the trained robot policy with the same user for 25 iterations (2500 timesteps, 150s of wall clock time). We then optimize for the prediction loss (including the KL regularization term) using a learning rate of 0.0001 for one to five steps and use the one with the lowest loss. We automatically tune the hyperparameters by using the training human distribution to collect a mini training set and evaluation set both of the same sizes as in test-time optimization. We use the mini training set to tune the learning rate and use the mini evaluation set to ensure there is no over-fitting.
>
> > Q: Co-optimization seems to be limiting in practice.
>
> Re: This is a great question. As we discussed in section 4.7, the co-optimization framework has its own limitations and, depending on the task, it may be more or less challenging to generate synthetic humans that capture real-world human behaviors. One interesting direction is to combine motion data to create synthetic human models, which capture suboptimal human behavior found in real life. Given such synthetic humans, we can then train the corresponding robot expert policies and carry out our framework as it is.
>
> We would also like to highlight that we now include the results of a user study with real human subjects in section 4.6 of our revised submission. Importantly, we find that robot policy trained with PALM achieves good zero-shot generalization performance when interacting with human participants. These results provide important evidence that the approaches we study in our paper have the potential to improve robot zero-shot and few-shot adaptation, despite the fact that there will always be sim2real gaps between synthetic humans and real humans at test-time.

---

### Official Review · Reviewer_QYB2 · 2022-08-01

**Originality:** Good
**Technical Quality:** Good
**Clarity Of Presentation:** Very Good
**Impact:** 3

**Recommendation:**

Weak Accept: I recommend accepting the paper, but will not argue for my recommendation if the majority of other reviewers have a different opinion.

**Summary:**

This paper tackles the problem of generalizing to out-of-distribution human behavior when using sim2real methods to develop assistive robotics capabilities. To that end, the authors formulate what they call the 'Assistive Personalization Problem': an assistive problem setting focused on generalization to unseen human behavior and preferences. To train policies for this problem setting, the authors propose the Prediction-based Assistive Latent eMbedding (PALM) framework, which involves learning a latent space by predicting future human actions from a history of observations and actions. Empirical results show that this approach outperforms state-of-the-art sim2real approaches used in non-assistive tasks and human-robot interaction approaches that train representations by predicting observed states or rewards. Additionally, the authors include a mechanism to adapt the learned latent space at test time and show empirical results that this helps with generalization.

**Issues:**

To improve the submission, I would suggest including another couple of assistive tasks to your empirical results or adding some discussion that makes a strong argument for why the experiments presented are expected to generalize to other tasks.

Because sim2real transfer is not really addressed in your experiments, you might also consider reframing the paper to have less emphasis on sim2real in the motivation.  Or you might consider including additional discussion to help readers conceptually bridge the gap between this work and the idea of sim2real transfer that motivates it.

**Quality Of The Limitations Section:**

Limitations are addressed clearly

**Reviewer Expertise:**

3: The reviewer is fairly confident that the evaluation is correct

**Robotics Focus:**

Highly relevant to robotics but no hardware experiments

**Strengths And Weaknesses:**

**Strengths**
- In general, I found the paper to be well written, easy to follow, and enjoyable to read
- The problem setting is well motivated and clearly described
- The proposed method of learning a latent space (including test-time optimization) is a natural and elegant approach to the problem
- The experiments and ablations are sensibly designed
- The empirical results presented in the paper are promising

**Scope for Improvement**
- The tasks used to evaluate the approach are relatively simple (reaching and itch scratching). I think the contribution could be stronger if the authors included additional, potentially more complex, assistive tasks in their empirical analysis (e.g. dressing, bathing, feeding, drinking).
- Sim2Real is used to motivate the work, however the work is done entirely in simulation with synthesized human behavior. It would strengthen the submission if the authors included some evaluation with realistic human preferences and behavior.

**Summary Of Recommendation:**

I am recommending to accept the paper because it presents a novel problem formulation that is interesting and well motivated. Additionally, the proposed learning-based approach to the problem is sensible and elegant -- and the empirical results presented in the paper are strong.

---

> ### Author Response · Authors · 2022-08-26
> **Moved user study to main text; working on additional tasks**
>
> > Q: The tasks used to evaluate the approach are relatively simple (reaching and itch scratching). I think the contribution could be stronger if the authors included additional, and potentially more complex, assistive tasks in their empirical analysis (e.g. dressing, bathing, feeding, drinking).
>
> Re: We thank the reviewer for this feedback. We are currently working on adding the bed bathing task to our revision. We would also like to note that the itch-scratching task we use has a 24-dimensional continuous observation space and 7-dimensional continuous action space for the robot. Prior work, e.g. LILI [17],  uses lower-dimensional tasks such as Lunar lander (8-dimensional continuous observation space with 4 discrete actions) and driving and air-hockey tasks (1-dimensional action spaces that change steering angle or lateral displacement). We believe that the high-dimensional continuous observation and action spaces in the itch scratching domain make this domain quite challenging compared to the benchmarks used in prior work, especially since the robot is required to not just reach the unobserved itch location, but also needs to apply only a small amount of pressure while moving back and forth in order to complete the task.
>
> > Q: Sim2Real is used to motivate the work, however, the work is done entirely in simulation with synthesized human behavior. It would strengthen the submission if the authors included some evaluation with realistic human preferences and behavior.
>
> Re: We would like to highlight the fact that we did run a user study with human subjects, but included it in the appendix due to page limit. We agree that these results are important and that they strengthen our paper. As such, we have moved them to the main text in section 4.6 of our revised submission. Importantly, we find that robot policy trained with PALM achieves good zero-shot generalization performance when interacting with real human participants.

---

### Meta-Review · Area_Chair_8zEk · 2022-08-09

**Recommendation:** Accept (Poster)
**Confidence:** 4

**Metareview:**

This paper considers the task of training policies for human-robot collaborative tasks and observes the need for out of distribution generalisation for human policies as a pre-requisite for training. The authors present an approach for learning a latent representation for human policies that may be adapted online.

Reviewers consider the motivation compelling and highly relevant for assistive robotics applications. In particular, reviewers consider the idea of learning a latent space for human policies as elegant. Further, thorough comparisons with related efforts strengthen the paper’s arguments.

The primary weakness of the paper is that the tasks used to evaluate the approach are relatively simple (reaching and itch scratching). This raises the question of whether and to what extent the method can scale to more realistic tasks. Any experiments to suggest that the approach can bridge the sim2real gap have been suggested since the current results are purely in simulation. A better technical exposition of the test time adaptation has also been suggested.

During the rebuttal phase,  the authors provided additional clarification/results for key reviewer questions such as mechanism for generalisation, overfitting, choice/training of encoders and their inclusion strengthens the overall paper. The major quantitative evaluation is on moderately complex tasks (e.g., itch scratching) supplemented by a user study. The user study is conducted in lab scenario (not with real users) and the setup bears resemblance to human studies in other human-collaborative domains (such as driving, collaborative assembly etc.).

Overall, the paper contributes an approach for learning latent representations from interaction data that can be adapted online based on distribution shifts observed online. The manuscript can be strengthened by a clearer exposition on the mechanism/nature of generalisation attained, mechanisms for preventing overfitting and and an analysis of potential scalability to complex scenarios. Further, since the learning problem is scoped towards the specific domain of assistive tasks (as opposed to general robot tasks), a clearer setup of the problem context and a statement on the scalability of the results to end-users may also be considered by the authors.